# Exploring the Feelings of Nurses during Resuscitation—A Cross-Sectional Study

**DOI:** 10.3390/healthcare10010005

**Published:** 2021-12-21

**Authors:** Anton Koželj, Maja Šikić Pogačar, Sabina Fijan, Maja Strauss, Vita Poštuvan, Matej Strnad

**Affiliations:** 1Faculty of Health Sciences, University of Maribor, 2000 Maribor, Slovenia; sabina.fijan@um.si (S.F.); maja.strauss@um.si (M.S.); 2Faculty of Medicine, University of Maribor, 2000 Maribor, Slovenia; maja.sikic@um.si (M.Š.P.); matej.strnad@um.si (M.S.); 3Slovene Center for Suicide Research, Andrej Marušič Institute, University of Primorska, 6000 Koper, Slovenia; vita.postuvan@upr.si; 4Emergency Department, University Clinical Centre Maribor, 2000 Maribor, Slovenia; 5Center for Emergency Medicine, Prehospital Unit, Community Healthcare Center, 2000 Maribor, Slovenia

**Keywords:** emergency nurses, resuscitation, stress, emotions

## Abstract

Cardiopulmonary resuscitation (CPR) is one of the most stressful situations in emergency medicine. Nurses involved in performing basic and advanced resuscitation procedures are therefore exposed to a certain amount of stress. The purpose of this study was to determine the stressors and the level of stress experienced by nurses during resuscitation. A cross-sectional quantitative study was done. The sample consisted of 457 nurses who worked in emergency units. First demographic data were collected, followed by a questionnaire regarding the effect of different situations that occur during and after resuscitation on nurses including Post-Code Stress Scale questionnaire. The most disturbing situations for respondents were resuscitation of young person (MV = 3.7, SD = 1.4), when they fail to establish an intravenous pathway (MV = 3.5, SD = 1.4), chaotic situation during resuscitation (MV = 3.4, SD = 1.4) and making decision about termination of resuscitation (MV = 3.1, SD = 1.5). Research has shown that nurses are exposed to a certain amount of stress during resuscitation, but most of them manage to compensate for stress effectively.

## 1. Introduction

Professional resuscitation guidelines are well known and presented in the form of algorithms. The latest guidelines for resuscitation have been created by the European Resuscitation Council (ERC) in 2021. However, even with knowledge of the guidelines, many decisions still need to be made in the clinical practice itself [1,2,3,4,5].

Cardiac arrest situations present one of the highest emergencies. In these cases, resuscitation must be started as soon as possible. The main goal is to restore cardiopulmonary function. Appropriate action in the event of a sudden cardiac arrest should be prompt and professional [6,7,8,9]. Cardiac arrest is the third leading cause of death in Europe [10,11]. The annual incidence of out of the hospital cardiac arrest in Europe is between 67 and 170 per 100,000 inhabitants. Resuscitation is attempted or continued by emergency medical service (EMS) personnel in about 50–60% of cases (between 19 and 97 per 100,000 inhabitants). The rate of bystander cardiopulmonary resuscitation (CPR) varies between and within countries (average 58%, range 13–83%). The annual incidence of intrahospital cardiac arrest is between 1.5 and 2.8 per 1000 hospital admissions. Survival rates at 30 days/hospital discharge range from 15% to 34% [5].

In the event of a cardiac arrest, in the hospital, the resuscitation team is immediately activated. In the absence of a physician, a nurse must take over the management of the resuscitation [4,12,13]. It is estimated that between 40% and 84% of all resuscitation attempts within critical care units result in the immediate or imminent death of the patient within 24 h [14,15,16]. Failure on a such large scale can therefore leave unpleasant psychological and professional consequences for nurses. During resuscitation, healthcare professionals can be exposed to several hazards. One of them is also a possibility to contract infections and diseases, e.g., COVID-19 [17,18,19,20,21]. Working in the prehospital environment and the emergency area is recognized as a high-frequency place for violence to occur. Nurses are more often exposed to violence than doctors, and female nurses suffer mental violence to a much higher extent than male nurses [22,23].

Nurses are generally the first responders to a cardiac arrest and initiate basic life support while waiting for the advanced cardiac life support team to arrive. The speed and competence of the first responder are determinants contributing to the initial survival of a person following a cardiac arrest [24,25,26,27]. The involvement of nurses in resuscitation cases can create a unique heightened level of psychological stress referred to as post-code stress, activation of coping behaviors and symptoms of posttraumatic stress disorder (PTSD). Because critical care nurses have frequent and cumulative exposures to unsuccessful cardiopulmonary resuscitations, psychological trauma often ensues. Demand for nurses in critical care and vacancy rates are high [28,29]. Situations are also stressful when it is necessary to decide whether to start, finish or abandon resuscitation. If a doctor is not present at the event, the decision must be made by a nurse [9].

This study aimed to present some aspects of nurses’ experiences during and after resuscitation procedures and to identify the stressors connected with resuscitation that nurses encounter and to what extent these stressors affect nurses.

## 2. Materials and Methods

### 2.1. Study Design and Participants

The present study utilized a cross-sectional observational design employing a survey method to collect information about demographics and psychological stress associated with CPR. The sample consisted of nurses and medical technicians with a national vocational qualification, who worked in emergency units (both hospital and prehospital). A smaller group of respondents had an additional level of education (master’s degree or specialization). Medical doctors were not included in the study. A total of 457 respondents were included in our study (95% confidence level) [30]. According to statistical data from the Slovenian National Institute of Public Health, there are 9043 nurses in Slovenia. The female to male ratio at the national level in Slovenia is 6.1 to 1 (87.3% female and 12.7% male nurses) [31].

### 2.2. Instruments

For the study, an anonymous questionnaire was used to assess the stress perceived by nurses who perform CPR. The first part of the questionnaire contained the demographic data of respondents and some individual questions. The second part included questions regarding the effect of different stress situations that occur during CPR and assessing the stress by measuring the magnitude of the psychological stress associated with CPR by using the Post-Code Stress Scale [32]. The Post-Code Stress Scale was specially designed for measuring stress among the population of nurses during and after resuscitation cases, and participants were asked to specify the extent to which each of the 20 items bothered them. Thus, higher scores on the scale indicated greater stress experienced by the nurses who were performing CPR. The first questionnaire used the 5-level Likert scale, where: 1—the claim does not apply to me, 2—the claim hardly applies to me, 3—I cannot decide whether the claim applies to me or not, 4—the claim often applies to me and 5—the claim is typical of me. Each item of the Post-Code Stress Scale questionnaire options ranged from 1 to 5 on the Likert scale, where: 1—it does not bother me at all, 2—it bothers me somewhat, 3—I am undecided or do not know if it bothers me, 4—it bothers me moderately and 5—it bothers me a lot.

The survey was carried out from March until December 2020 at the University Medical Centre Maribor, 20 prehospital emergency units in Slovenia, the emergency medical dispatch center (EMDC), and the Slovenian Army Medical Service. Signed informed consents from institutions were collected before the institutions joined the study. The questionnaires were distributed through the heads of the department. Additional explanations were provided if necessary.

### 2.3. Scale Validity

To estimate the internal consistency reliability of the questionnaire, Cronbach’s alpha (α) was calculated. Reliability testing with a sample of nurses demonstrated good internal consistency for the part of the questionnaire covering the effect of specific situations that arise during CPR on nurses, Cronbach’s α = 0.840 (18 items). The internal consistency reliability for the part of the questionnaire covering the stress following CPR (19 items) and for the Post-Code Stress Scale (20 items) was also good (Cronbach’s α = 0.936 and 0.939, respectively).

### 2.4. Ethical Considerations

Ethical approval was obtained through the National Medical Ethics Committee of Slovenia by the Ministry of Health (reference number: 0120-517/2017/5).

### 2.5. Statistical Analysis

Descriptive statistics were used to characterize participants’ demographic details. Data were presented by frequencies and percentages for categorical variables, or by means and standard deviations for continuous variables. T-test for independent samples, Pearson correlation test, and ordinal logistic regression were used to identify associations between the variables. The statistical analyses were conducted using IBM SPSS ver. 27.0 (IBM Corp., Armonk, NY, USA). Statistical significance was set at *p* < 0.05.

## 3. Results

### 3.1. Sample Description

The sample was nonrandom and purposeful. The study involved 457 nurses, of which 292 (63.9%) were male and 165 (36.1%) were female. The survey included health professionals of different ages and different numbers of years and work experience in health care. (Table 1). Research included nursing staff employed in prehospital (*n* = 290, 63.3%), hospital (emergency departments and intensive care units) (*n* = 147, 32.1%) and other areas (EMDC and Slovenian Army Medical Service; *n* = 20, 4.6%). In Slovenia, nurses work in ambulance vehicles. More than half of the participants were nurses (*n* = 283, 62.0%), and 135 of respondents (29.5%) were medical technicians with a national vocational qualification which allows them almost the same competencies as nurses have to work in a prehospital setting. Thirty-seven (8.1%) respondents had an additional level of education (master’s degree or specialization), and two (0.4%) participants did not specify their education level. 

Most of the respondents (203, 44.4%) were involved in one to five resuscitations per year in the presence of a physician (Table 2). One hundred thirty-three (29.1%) of them participated in six to ten resuscitations per year while 31 (6.8%) of the respondents participated in >20 resuscitations per year in the presence of the physician. The number of respondents involved in resuscitations in absence of physicians was also assessed. 

Our results showed that almost half of the respondents (*n* = 223, 48.8%) were involved in one to five resuscitations per year in absence of a physician. Many respondents (*n* = 203, 44.4%) never participated in resuscitations without a physician. On the other hand, only a few of them participated in 16 to 20 or more than 20 resuscitations per year without a physician being present (0.4% in both cases).

### 3.2. Education and Experience with Resuscitation

Nurses involved in the present study attended different resuscitation courses (Figure 1). The majority (73.0%) of respondents had completed internal courses in resuscitation. In addition, 53.7% of respondents had completed courses in resuscitation held by *The Nurses and Midwives Association of Slovenia*, which is the national regulatory authority for the field of nursing and midwifery in Slovenia. To work in a prehospital environment in Slovenia, medical technicians must complete additional training and obtain a national vocational qualification. At the same time, both medical technicians and nurses have to complete the knowledge and skills evaluation to work in the prehospital Emergency Medical Service in Slovenia following Rules on EMS. The validity of the knowledge test is five years. Nurses and medical technicians do not need additional qualifications to work in the emergency departments of hospitals. However, while working, it is recommended that they take part in certain international courses, which have been successfully operating in Slovenia for many years. Approximately a third of respondents completed the Advanced Life Support (30.6%), International Trauma Life Support (29.2%), or Medical Response to Major Incidents (27.0%). Only 5.7% completed Immediate Life Support and only 6.5% of respondents completed European Paediatrics Advanced Life Support. About 6.3% of respondents declared having completed other resuscitation courses.

Most of the participants (66.7%) in the present study declared they knew the algorithm well enough to perform basic resuscitation procedures; however, 24.4% of them expressed their wish to acquire additional knowledge or renew their knowledge in this field.

Feelings of respondents during the resuscitation and their experiences with the resuscitation in the presence or absence of a physician were observed (Table 3). Results showed that 61.2% of respondents felt their experience of resuscitation in the presence of a physician depended on the individual physician’s knowledge and ability to lead a team. Next, 20.9% of respondents found it easier to assist in resuscitation when a physician was present, and 13.9% of them declared the presence or absence of a physician did not affect their experience of resuscitation.

### 3.3. Analysis of Questionnaires

Respondents were asked about their feelings during the resuscitation. The respondents rated 18 statements on a five-point scale, which referred to their feelings during a resuscitation procedure, with scores ranging from 1 to 5 similar to Post-Code Stress Scale. Procedures that were most disturbing for respondents were artificial mouth-to-mouth ventilation (MV = 3.6, SD = 1.4), when they fail to establish an intravenous pathway (MV = 3.5, SD = 1.4), chaotic situation during resuscitation (MV = 3.4, SD = 1.4) and making a decision about the termination of resuscitation (MV = 3.1, SD = 1.5). During resuscitation procedures, conditions that bother the respondents the least were the preparation and application of medicines ordered by a physician (MV = 1.4, SD = 0.9), chest compressions (MV = 1.4, SD = 0.9), calling an additional team for help (MV = 1.4, SD = 0.8), artificial ventilation with an Ambu bag (MV = 1.4, SD = 0.9), determining the absence of vital signs (MV = 1.7, SD = 1.2) and the decision to start resuscitation (MV = 1.7, SD = 1.1). Slightly more disturbing for the respondents was when an injury or fracture of the chest occurred during resuscitation (MV = 2.0, SD = 1.2) or when resuscitation took place just because of the possibility of organ donation (MV = 2.0, SD = 1.3).

Respondents’ feelings after resuscitations were also assessed. Again, the respondents rated 19 statements on a five-point scale, which referred to their well-being following a resuscitation procedure. According to individual claims about their well-being after resuscitation, the respondents stated for the following claims that they agree with them the least which means that resuscitation had the least effect on them: “Due to participation in resuscitation procedures, I spend less time with my friends” (MV = 1.2, SD = 0.6), “Due to participation in resuscitation procedures, I partially neglect my family members” (MV = 1.2, SD = 0.6), I consume more alcohol than usual after work with resuscitation (MV = 1.3, SD = 0.8), “After resuscitation, I smoke more cigarettes than usual” (MV = 1.4, SD = 0.9), “I notice that after difficult resuscitations I am more conflicted in the home environment than usual” (MV = 1.4, SD = 0.9) and “Due to participation in resuscitation procedures, I run out of motivation and strength for recreational activities” (MV = 1.4, SD = 0.8). The highest score was given for the following statements: “After resuscitation I am shaken” (MV = 2.3, SD = 1.1), “I am still thinking about resuscitation” (MV = 2.3, SD = 1.2) and “After resuscitation I feel physically more tired” (PV = 2.4; SO = 1.2). In this, the last set of claims, the respondents did not rate any of the claims with an average value of 3 or more which means that they did not strongly agree with the claims.

The Post-Code Stress Scale was specifically designed for measuring stress among nurses and medical technicians (nonphysician medical staff) during and after resuscitation [32]. Results are presented in Table 4.

Eleven of all twenty items on the scale (1, 3, 5, 7, 9, 10, 14, 15, 17, 19 and 20) reflect internal sources of stress. The remaining nine items (2, 4, 6, 8, 11, 12, 13, 16 and 18) reflect external sources of stress and are validated as such. As it can be seen from Table 4, the respondents had the least problem with the following claims: “When my peers are quick to notice and point out that I made a mistake” (MV = 2.3, SD = 1.3) and “When my hands shake during a code” (MV = 2.4, SD = 1.3).

The following claims were most disturbing for respondents: “When I code someone young” (MV = 3.7, SD = 1.4), “When I am unable to make a properly functioning piece of equipment operate during a code” (MV = 3.4, SD = 1.4), “When a nurse manager/supervisor criticizes me when I’ve done my best” (MV = 3.4, SD = 1.4), “When a nurse manager doesn’t provide assistance during a code” (MV = 3.4, SD = 1.3) and “When I think I might have missed a sign or symptom that would have helped me predict that the patient would code” (MV = 3.3, SD = 1.3).

Afterward, items from the 20-item scale were grouped into five dimensions according to different sources of stress [32] (Table 5). The dimension I indicated that the source of stress was from feeling discomposed. Items in dimension II revealed that the source of stress was from feeling oppressed. In dimension III, the items implied that the source of stress was from feeling uncertain. Furthermore, the source of stress in dimension IV resulted from feeling burdened, and the items in dimension V indicated that the source of stress was from feeling morally conflicted. Overall, higher scores reflected greater amounts of stress [32].

Stress from feeling discomposed (MV = 11.2, SD = 3.8) represented the highest source of stress among respondents in our study. Stress from feeling burdened was less represented as a source of stress (MV = 9.5, SD = 3.3). The least important source of stress was stress from feeling morally conflicted (MV = 6.0, SD = 2.4) (Table 5).

Finally, associations between participants’ characteristics (i.e., years of service in healthcare, sex, area of employment, the approximate number of resuscitations with/without physician supervision) and each of the five dimensions from the Post-Code Stress Scale are presented in Table 6.

The results in Table 6 show that respondents with more years in healthcare service experienced less stress from feeling discomposed (OR = 0.98, 95% CI = 0.96–1.00, *p* = 0.040). In addition, male nursing practitioners experienced less stress from feeling discomposed (OR = 0.56, 95% CI = 0.37–0.85, *p* = 0.006). Male nursing practitioners experienced less stress that arose from feeling oppressed (OR = 0.6, 95% CI = 0.40–0.90, *p* = 0.012) while health care practitioners employed at the hospital experienced more stress from feeling oppressed when compared to others (OR = 3.11, 95% CI = 1.30–7.48, *p* = 0.012). Furthermore, male nursing practitioners experienced less stress that arose from feeling uncertain (OR = 0.45, 95% CI = 0.30–0.68, *p* < 0.001), from feeling burdened (OR = 0.45, 95% CI = 0.30–0.68, *p* < 0.001) or from feeling morally conflicted (OR = 0.57, 95% CI = 0.38–0.86, *p* = 0.008).

## 4. Discussion

In our study, the experience of nurses in one of the most stressful healthcare situations was investigated with a focus on their experiences and feelings during and after a resuscitation intervention. Most nurses were involved in one to five resuscitations annually (44.4% with physician being present, 48.7% without physician being present). The second largest group was involved in 6 to 10 cases annually (29.1%) with the physician being present, and 5% without the physician being present. The difference in the experience of resuscitation of nurses in the presence of a physician was compared, and it was found that most of the respondents (61.2%) claimed that the difference in presence of the physician depended on the physician’s knowledge and ability to lead the team. One study by Arshid et al. [33] also showed that the number of mistakes made during resuscitation depends on the ability of the team leader. In a simulation environment [34], team leaders often overlooked the misconduct of other team members and did not correct the resulting errors. Teams led by emergency medicine residents re-established return of spontaneous circulation (ROSC) faster, but they exhibited no differences in survival rates. One study showed that training improved leadership, coordination behavior and BLS skills. Leadership performance and experience influence team performance [35]. Leadership must be trained specifically. Role awareness and consideration of others’ skills have a positive influence on team behavior. Conflict is detrimental. Communication affects cohesion [36]. The adherence to the ALS algorithm is significantly higher in teams in which a command physician was identified [37]. A prolonged process of team building and poor leadership are associated with significant shortcomings in CPR [38]. Effective leaders play a pivotal role in promoting team performance. Leadership behavior as a factor contributing to effective CPR performance has both positive and negative effects [39]. It has been shown that survival after out-of-hospital cardiac arrest was not different for patients treated by the team with the physician or the team without the physician [40]. In the prehospital setting, it can be difficult to obtain additional help from a resuscitation team. However, modern telecommunication methods are already being introduced (e.g., real-time video image applications) that enable communication between the prehospital and hospital environments, between nurses, paramedics and doctors [41,42,43].

Many countries around the world have well-regulated clinical competencies of rescuers [44,45,46]. In Slovenia, the clinical competencies of nurses during advanced life support are not well defined. That situation can cause additional stress for them.

The questionnaire developed by Cole, Slocumb and Mastey includes five Post-Code Stress Scale dimensions. Cole, Slocumb and Mastey [32] found the highest score range for feeling discomposed (mean value 15.1) and the lowest (mean value 8.2) for feeling uncertain and for feeling the moral conflict. Similarly, our results found the highest score for feeling discomposed (mean value 11.2) and the lowest for feeling morally conflicted (mean value 6.0). Comparable results were also found for feeling oppressed, 8.8 reported by Cole et al. [32] and 8.6 for the present study, as well as for feeling uncertain, 8.2 reported by Cole et al. [32] and 8.5 for the present study. Cole et al. [32] also concluded that the Post-Code Stress Scale shows promise as a measure of stress experienced by critical care nurses, as well as the other nurses, who do not typically participate in resuscitation attempts. Nurses working in critical care areas such as intensive care units and emergency departments have greater exposure to resuscitations; however, working in these units has been linked to greater levels of burnout, compassion fatigue and post-traumatic symptoms [47]. A statistically significant difference in feeling oppressed among hospital workers compared to the prehospital workers or others (EMDC and Slovenian Army Medical Service) (*p* = 0.012) was found in the present study. Hu et al. [48] found that 71.3% of the responding doctors and 68.3% of the responding nurses were deemed to be burnt out. People working in the general ICU were most likely to be burnt out [49]. Another study showed the prevalence of burnout syndrome is 55.3% of ICU nurses [50].

Critical care nurses experience stress from multiple sources. One source of stress may arise from participation in resuscitation attempts, and this has been labeled as post-code stress. Post-code stress is posited to initiate processes for regulating emotions, such as coping behaviors, that aim to maintain an individual’s psychological health and prevent the manifestation of stress-associated symptoms [29,32].

Family presence during resuscitation can also present additional stress. This can be both positive and negative. In any case, it requires additional commitment, skills and effort from the nurse. However, it can also result in emotional responses in nurses that can last for extended periods. In general, family presence does not worsen resuscitation outcomes, but it can improve outcomes for family members [51,52,53,54]. In the study by Porter and co-authors [55], the majority of nurses (83% of respondents) agreed with the presence of relatives during resuscitation and believed that relatives had the right to be present (79% of respondents). Moreover, 92% of respondents thought that such support for relatives is part of the scope of a nurse’s job. However, this represents additional effort and stress for them. During a child’s resuscitation, parents perceived a sense of overwhelming chaos, yet still had an innate need to be present and know what was going on. While emotional support was appreciated, the most important was to receive real-time clinical information from healthcare staff and to see and feel that the team was personally invested in their child [56]. In our study, we found that family presence caused medium stress levels (medium value 2.8). The lowest stress level (medium value 2.3) was caused by peers noticing and pointing out respondents’ mistakes and hands shaking during a code. The highest stress level was caused by coding some young (medium value 3.7) and being criticized by a supervisor (medium value 3.4) or not receiving help from a nurse manager (medium value 3.4) during resuscitation. Of the expected stressful situations (e.g., unsuccessful resuscitation, etc.), some items could be eliminated with better work organization and more appropriate team relationships [57,58,59]. Physicians and nurses had similar problems during resuscitation. Among the most frequently exposed problems, they pointed out lack of equipment, inadequate teamwork and communication and inadequate incident reporting [60].

The literature recognizes that resuscitation efforts are highly stressful events for all health professionals, but only a little information is known about how nurses experience these situations [25]. Cardiopulmonary resuscitation (CPR) causes significant stress for the rescuers which may cause deficiencies in attention and increase distractibility. This may lead to misjudgments of priorities and delays in CPR performance, which may further increase mental stress (vicious cycle) [61].

Considering all findings, further research could include a review into the experience and feelings of nurses with the use of qualitative instruments such as interviews to explore an in-depth and extensive understanding of the complex problem of cooperating in a resuscitation. An important aspect for further research is to address and find efficient measures for preventing post-code stress of the complete resuscitation team.

## 5. Conclusions

Our research did not find a serious burden on the nurses; however, certain situations caused higher stress levels (resuscitation of children, equipment failure, unfounded criticism from the leader, etc.) than others. We believe that most of these stressful situations for nurses could be reduced or even eliminated with better work organization, a healthy work environment, regular debriefing after resuscitation and additional education. In the debriefing, all team members should be involved and speak openly about their resuscitation experience. Exposed irregularities should not be seen as criticism but as opportunities to correct mistakes in the future. Resuscitation team members should consider the destress techniques that work best for them, such as stress management training, mediation, awareness-raising about nontechnical skills, cognitive aid, recognizing and reducing physiologic stress (heart rate, breathing, the impact of stress hormones, etc.) and team support. Good, determined, professional and calm team management also plays an important role in reducing stress during resuscitation. Team members need to know exactly what is going on during resuscitation and what their job is. This has been shown to reduce stress. The division of labor, however, is the work of the team leader. Leadership must be trained specifically. We also propose the establishment of predetermined routes, persons or procedures in each institution that can be used by people who assess that they are under stress (confidential persons, superiors, psychologists, or lawyers). However, individuals should also use their techniques such as sports and cultural activities; conversation with colleagues, friends and partners; and meditation to help them cope with stress.

## Figures and Tables

**Figure 1 healthcare-10-00005-f001:**
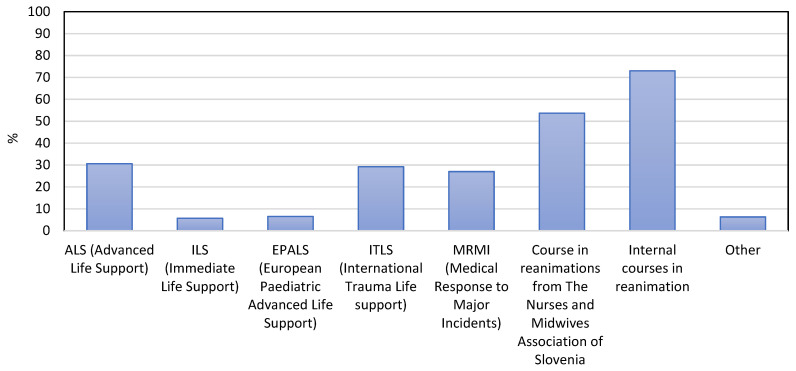
Different courses in resuscitation were completed by the nurses included in the study.

**Table 1 healthcare-10-00005-t001:** Age (years) and years of work experience and practice in nursing.

Age and Work Experience	Min	Max	MV
Age	20	61	36.2
Years of service in healthcare	1	41	13.8
Years of service in the emergency area	1	40	10.6

Legend: min = minimum, max = maximum, MV = mean value, SD = standard deviation.

**Table 2 healthcare-10-00005-t002:** Approximated number of resuscitations in which nurses participate per year.

Approximated Number of Resuscitations per Year	With a Physician		Without Physician	
	*n*	%	*n*	%
None	25	5.5	203	44.4
1–5	203	44.4	223	48.8
6–10	133	29.1	23	5.0
11–15	43	9.4	4	0.9
16–20	22	4.8	2	0.4
>20	31	6.8	2	0.4

Legend: *n* = number of responses, % = percentage.

**Table 3 healthcare-10-00005-t003:** The difference in the experience of resuscitation with or without the presence of a physician as a team leader.

Experiencing Resuscitation with or without the Presence of a Physician	*n*	%
The presence or absence of a physician does not affect my experience of resuscitation	63	13.9
I find resuscitation easier when a physician is present	95	20.9
I find resuscitation more difficult when a physician is present	10	2.2
It depends on the physician’s knowledge and ability to lead a team	278	61.2
Without the presence of a physician, resuscitation is not performed	8	1.8

Legend: *n* = number of responses, % = percentage.

**Table 4 healthcare-10-00005-t004:** Feelings of respondents during resuscitation (Post-Code Stress Scale directions and Items).

No.	Items	MV	SD
1	When my hands shake during a code	2.4	1.3
2	When my peers are quick to notice and point out that I made a mistake	2.3	1.3
3	When I feel like I didn’t function well during a code	3.2	1.3
4	When more than one doctor gives orders during a code	3.2	1.4
5	When I lose my confidence during a code	2.8	1.3
6	When a patient’s family thinks I can keep him/her alive	2.8	1.3
7	When I am unable to make a properly functioning piece of equipment operate during a code	3.4	1.4
8	When hospital policies/procedures are conflicting	3.2	1.4
9	When we code a patient, I believe we should not code	3.0	1.4
10	When I have trouble reading the ECG	3.0	1.3
11	When I code some patients only because hospital policy says I must	3.0	1.4
12	When a nurse manager/supervisor criticizes me when I’ve done my best	3.4	1.4
13	When a nurse manager doesn’t provide assistance during a code	3.4	1.3
14	When I wonder if I made a mistake	2.9	1.2
15	When I am not permitted time to regroup and pick myself up after a code	3.0	1.4
16	When people think I can function appropriately immediately after a code	2.8	1.4
17	When I code someone young	3.7	1.4
18	When no one talks about the code after it is over	2.7	1.4
19	When I think I might have missed a sign or symptom that would have helped me predict that the patient would code	3.3	1.3
20	When the patient dies	2.8	1.3

Legend: MV = medium value, SD = standard deviation.

**Table 5 healthcare-10-00005-t005:** Items of Post-Code Stress Scale [32] grouped into five dimensions.

Source of Stress	Min	Max	MV	SD
From feeling discomposed	4	20	11.2	3.8
From feeling oppressed	3	15	8.6	3.4
From feeling uncertain	3	10	8.5	3.0
From feeling burdened	3	15	9.5	3.3
From feeling morally conflicted	2	10	6.0	2.4

Legend: min = minimum, max = maximum, MV = medium value, SD = standard deviation.

**Table 6 healthcare-10-00005-t006:** Multivariable associations between participant characteristics and stress that arises from feeling discomposed, oppressed, uncertain, burdened and morally conflicted.

Source of Stress	Discomposed		Oppressed		Uncertain		Burdened		Moral Conflict	
	OR (95% CI)	*p*	OR (95% CI)	*p*	OR (95% CI)	*p*	OR (95% CI)	*p*	OR (95% CI)	*p*
Years of service in healthcare										
Years of service	0.98 (0.96–1.00)	0.040	1.01 (0.99–1.03)	0.311	0.98 (0.96–1.00)	0.053	1.01 (0.99–1.02)	0.424	1.01 (0.99–1.03)	0.251
Sex										
Female	1.00 (reference)		1.00 (reference)		1.00 (reference)		1.00 (reference)		1.00 (reference)	
Male	0.56 (0.37–0.85)	0.006	0.60 (0.40–0.90)	0.012	0.45 (0.30–0.68)	<0.001	0.45 (0.30–0.68)	<0.001	0.57 (0.38–0.86)	0.008
Area of employment										
Other *	1.00 (reference)		1.00 (reference)		1.00 (reference)		1.00 (reference)		1.00 (reference)	
Prehospital	1.16 (0.51–2.68)	0.727	1.49 (0.65–3.44)	0.361	1.13 (0.49–2.61)	0.802	0.79 (0.34–1.83)	0.575	1.44 (0.62–3.33)	0.387
Hospital	1.81 (0.76–4.32)	0.203	3.11 (1.30–7.48)	0.012	2.18 (0.91–5.21)	0.097	1.09 (0.46–2.60)	0.885	2.10 (0.87–5.06)	0.096
The approximate number of resuscitations per year with physician supervision										
0	1.00 (reference)		1.00 (reference)		1.00 (reference)		1.00 (reference)		1.00 (reference)	
1–5	0.75 (0.34–1.63)	0.555	0.66 (0.30–1.43)	0.391	1.14 (0.52–2.49)	0.566	1.04 (0.48–2.27)	0.790	0.99 (0.45–2.17)	0.961
6–10	0.65 (0.29–1.48)	0.384	0.54 (0.24–1.22)	0.215	0.89 (0.39–2.02)	0.996	1.05 (0.46–2.37)	0.761	0.93 (0.41–2.10)	0.843
11–15	0.77 (0.30–1.96)	0.668	0.91 (0.36–2.32)	0.966	1.48 (0.58–3.76)	0.308	1.43 (0.56–3.64)	0.364	1.71 (0.67–4.37)	0.264
16–20	0.72 (0.24–2.14)	0.479	0.87 (0.29–2.57)	0.714	1.00 (0.34–2.96)	0.908	1.94 (0.65–5.76)	0.342	2.07 (0.69–6.21)	0.318
>20	0.66 (0.25–1.77)	0.503	0.42 (0.16–1.13)	0.117	0.94 (0.35–2.52)	0.872	1.09 (0.40–2.92)	0.745	0.96 (0.35–2.59)	0.900
The approximate number of resuscitations per year without physician supervision										
0	1.00 (reference)		1.00 (reference)		1.00 (reference)		1.00 (reference)		1.00 (reference)	
1–5	1.01 (0.69–1.46)	0.997	1.29 (0.89–1.88)	0.190	1.28 (0.88–1.86)	0.197	1.08 (0.74–1.56)	0.699	1.43 (0.98–2.09)	0.061
>6	0.85 (0.40–1.80)	0.318	1.02 (0.48–2.16)	0.390	0.74 (0.35–1.58)	0.437	0.76 (0.36–1.61)	0.467	1.17 (0.55–2.50)	0.684

Legend: OR: odds ratio, 95% CI: 95% confidence, * other: EMDC and Slovenian Army Medical Service.

## Data Availability

The study did not report any data.

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
