# Peer review of "Exploring the Feelings of Nurses during Resuscitation—A Cross-Sectional Study"

_healthcare, 2021, doi:10.3390/healthcare10010005_

Round 1
Reviewer 1 Report
An interesting paper on measuring stress levels of professionals during resuscitation.
Some additional information would benefit the paper.
The authors refer tot he total number of nurses in Slovenia. How many nurses are employed in hospital and prehospital emergency care? To give an analogy of the study sample size.
Referring to the male/female ratio in the study sample, please refer to the male/female nurse ratio at the national level.
The authors describe the study participants as nurses. Later in the text (Results section) it is stated "More than half of 125 the participants were nurses (n = 283 and 62.0%)". This creates confusion, please clarify the composition of the sample in the Methodology section.
Do the authors have information on the outcome (successful or not) of the resuscitation? Has this parameter been considered in the ascertainment of the post traumatic stress levels?
Line 205 Post-Code Stress Scale was used to determine post-code stress among nurses: does this mean only the nurses in the sample or the whole sample? If only the nurses why so?
Is it compulsory for a nurse to complete courses in resuscitation in order to work in emergency care?
Please also describe how a nurse is allocated in an emergency care department in Slovenia. Are there any specific criteria?
In the Conclusions section, the authors could also specify education on de-stress techniques along with debriefing sessions.
Some English editing is required, e.g. line 288 includes instead of include.
How exactly were the study participants approached. In the Results sections there is a phrase: the sample was not random and purposeful. Please explain.
Author Response
Reviewer 1 :
An interesting paper on measuring stress levels of professionals during resuscitation.
Some additional information would benefit the paper.
A
The authors refer to he total number of nurses in Slovenia. How many nurses are employed in hospital and prehospital emergency care? To give an analogy of the study sample size.
Author comment: Thank you for this comment. Unfortunately, the number of nurses working in the pre-hospital environment for Slovenia is not available. National Institute of Public Health which provides official and updated data on the number of all health care workers in Slovenia, does not provide this information separately, while the Rescue Section of Slovenia is a voluntary professional association. Membership is not mandatory, so their data of the number of nurses employed in the pre-hospital environment does not reflect the real picture, because not everyone is, of course, a member of the association. Therefore, unfortunately, valid information is not available in Slovenia.
Referring to the male/female ratio in the study sample, please refer to the male/female nurse ratio at the national level.
Author comment: Thank you for this comment. We inserted the data in the text.
The authors describe the study participants as nurses. Later in the text (Results section), it is stated "More than half of 125 the participants were nurses (n = 283 and 62.0%)". This creates confusion, please clarify the composition of the sample in the Methodology section.
Author comment: Thank you for this comment. We inserted the data in the text.
Do the authors have information on the outcome (successful or not) of the resuscitation? Has this parameter been considered in the ascertainment of the post-traumatic stress levels?
Author comment: Thank you for this comment. This study presented in this article did not cover the impact of the outcome of resuscitation procedures on respondents. It covered the experience and participation of respondents in resuscitation procedures in general. The only one / last question in Post-code Stress Scale is “When the patient dies”. The results are presented in the text, but interestingly, this claim did not prove to be very burdensome. But your comment confirms our assumption that the outcome of resuscitation is important, and we will investigate such effects in an additional “qualitative study”, where we will further investigate the impact of resuscitation outcome on respondents' (interviews). The research itself is already underway.
Line 205 Post-Code Stress Scale was used to determine post-code stress among nurses: does this mean only the nurses in the sample or the whole sample? If only the nurses why so?
Author comment: Thank you for this comment. Post-Code Stress Scale was used to determine post-code stress among nurses”. This is the original author's claim. (Differences in naming and staff structure in USA and Slovenia). However, in our study, this covers all surveyed nurses and medical technicians. We have further highlighted this in the text.
Is it compulsory for a nurse to complete courses in resuscitation in order to work in emergency care?
Author comment: Thank you for this comment. Nurses and medical technicians do not need additional qualifications to work in the emergency departments of hospitals. Completed education (for nurses or medical technicians) is enough. However, while working, it is recommended that they take part in certain international courses, which have been successfully operating in Slovenia for many years. In Slovenia, we perform international courses ALS (Advanced Life support), ILS (Immediate life support), EPALS (European Paediatric Advanced Life Support), ITLS (International Trauma Life Support), (I am an international instructor on these courses too), MRMI ( Medical Response to Major Incidents), ATCN (Advanced Trauma Care for Nurses). However, there are additional shorter courses conducted by certain institutions, the Chamber of Nursing of Slovenia, etc. The courses presented are highly recommended but are still optional for the time being. To work in a pre-hospital environment, medical technicians must complete additional training and obtain a national vocational qualification. At the same time, both medical technicians and nurses have to do complete the knowledge and skills evaluation for work in pre-hospital Emergency Medical Service in Slovenia in accordance to Rules on EMS (Official Gazette of Republic Slovenia, No. 81/15 (30. 10. 2015). The validity of the knowledge test is five years.
Please also describe how a nurse is allocated in an emergency care department in Slovenia. Are there any specific criteria?
Author comment: Thank you for this comment. In both cases (hospital and pre-hospital), no one is assigned anywhere. When the need arises for employment in such areas, a public tender is published and those who are interested in this type of work apply. In the case of a large number of candidates, a selection of candidates shall be made. (Preference is given to candidates who have already had some experiences and passed some of the above-mentioned international exams).
PS: We will insert this described explanation in abbreviated form into the text within the article.
PS2: There is still a great interest in work in pre-hospital units in Slovenia, (especially men), and there are always enough candidates. (But In general, there is a shortage of nurses in Slovenia).
In the Conclusions section, the authors could also specify education on de-stress techniques along with debriefing sessions.
Author comment: Thank you for this comment. The proposal is taken into account and included in the text.
Some English editing is required, e.g. line 288 includes instead of include.
Author comment: Thank you for this comment. Our native English speaker will go through a text again.
How exactly were the study participants approached. In the Results sections there is a phrase: the sample was not random and purposeful. Please explain.
Author comment: Thank you for this comment.
A non-random and purposeful sample means that we have precisely defined the criteria for who can participate in the research (nurses and medical technicians working in the field of emergency medicine). Questionnaires were delivered exclusively to hospital wards where they treat emergency patients and pre-hospital units. Thus, we purposefully included nurses and medical technicians who actually deal with emergencies and exclude from the survey other health professionals who do not regularly encounter emergencies in their work.

Reviewer 2 Report
The authors of the topic Feelings of Nurses During Resuscitation should be welcomed. It is an important and constantly relevant issue in terms of both cognitive and application. The intention of the research is correct, the stated goal is clear, but in order to increase the substantive value of the article, the reviewer proposes to find out why the age and level of education and experience in resuscitation were adopted as independent, explanatory variables and omitted, as indicated in the literature on the subject, e.g. personality conditions. This assumption would certainly be important when constructing a research tool. Another comment concerns the extension of the application aspects of the conclusions, which, in the opinion of the reviewer, were too synthetically formulated. Specific practices, e.g. workshop classes, should be indicated, which could expand the possibilities of coping with stress among nurses.
Author Response
Reviewer 2:
The authors of the topic Feelings of Nurses During Resuscitation should be welcomed. It is an important and constantly relevant issue in terms of both cognitive and application.
The intention of the research is correct, the stated goal is clear, but in order to increase the substantive value of the article, the reviewer proposes to find out why the age and level of education and experience in resuscitation were adopted as independent, explanatory variables and omitted, as indicated in the literature on the subject, e.g. personality conditions. This assumption would certainly be important when constructing a research tool.
Author comment: Thank you for this comment. The post-Code Stress Scale questionnaire used as the lead questionnaire in this study does not cover the study of personality conditions. While studying the literature, we determined the possible influence of gender, age, and experience, as well as the influence of personality conditions on team leadership. Authors Streiff with her coworkers e.g. finds out that leadership is determined by gender and personality and not by knowledge or experience. As you mentioned, consideration of personality conditions would help develop further research tools. Your proposal is very valuable to us, as we are continuing the research in which we will conduct interviews (qualitative study). A pilot study has already been done. In it, we have already taken into account some personal characteristics of the interviewees. However, we plann to expand on this topic in a follow-up study by taking into account some personality conditions such as extroversion, openness, agreeableness, and conscientiousness. Your comment thus confirms the fact that we are on the right track.
Another comment concerns the extension of the application aspects of the conclusions, which, in the opinion of the reviewer, were too synthetically formulated. Specific practices, e.g. workshop classes, should be indicated, which could expand the possibilities of coping with stress among nurses.
Author comment: Thank you for this comment. We have concluded with some examples of special methods that may be suitable for reducing stress in nurses. The conclusion included the following text: In the debriefing, all team members should be involved and speak openly in their resuscitation experience. Exposed irregularities should not be seen as criticism but as opportunities to correct mistakes in the future. Resuscitation team members should consider the de-stress techniques that work best for them such as, stress management training, mediation, awareness-raising about non-technical skills, cognitive aid, recognizing and reducing physiologic stress (heart rate, breathing, the impact of stress hormones, etc), and team support. Good, determined, professional, and calm team management also plays an important role in reducing stress during resuscitation. Team members need to know exactly what is going on during resuscitation and what their job is. This has been shown to reduce stress. The division of labor, however, is the work of the team leader. Leadership must be trained specifically. We also propose the establishment of predetermined routes, persons, or procedures in each institution that can be used by people who assess that they are under stress (confidential persons, superiors, psychologists, or lawyers. However, individuals should also use their techniques to help them cope with stress such as sports and cultural activities, conversation with colleagues, friends, partners, meditation, etc.
I hope we have understood and placed your comment accordingly.
